# Concerning the Etiology of Syrah Decline: A Fresh Perspective on an Old and Complex Issue Facing the Global Grape and Wine Industry

**DOI:** 10.3390/v15010023

**Published:** 2022-12-21

**Authors:** Huogen Xiao, Olivia Roscow, Julia Hooker, Caihong Li, Hans J. Maree, Baozhong Meng

**Affiliations:** 1Department of Molecular and Cellular Biology, the University of Guelph, Guelph, ON N1G 2W1, Canada; 2Department of Genetics, Stellenbosch University, Private Bag X1, Matieland 7602, South Africa

**Keywords:** grapevine, high-throughput sequencing, PCR, RT-PCR, Syrah, anthocyanin, drought, grapevine rupestris stem pitting-associated virus, rugose wood complex, graft incompatibility

## Abstract

Syrah decline, first identified in Southern France in the 1990s, has become a major concern in the global grape and wine industry. This disease mainly affects Syrah (Shiraz) grapevines. Characteristic symptoms include the bright and uniform reddening of leaves throughout the canopy in late summer or early fall; the appearance of abnormalities on the trunk, mainly at the graft union (swelling, pits, grooves, and necrosis); and a reduction in vine vigor, yield and berry quality. Diseased vines may die a few years after disease onset. Damages to the vine are even more pronounced in cool climate regions such as Ontario (Canada), where the affected vines are subjected to very cold and prolonged winters, leading to large numbers of vine deaths. Despite the extensive efforts of the global grape research community over the past few decades, the etiology of this disease remains unclear. In this study, we conducted extensive analyses of viruses in declining Syrah vines identified in commercial vineyards in the Niagara region (Ontario, Canada) through high-throughput sequencing, PCR, RT-PCR and the profiling of genetic variants of select viruses. Multiple viruses and viral strains, as well as three viroids, were identified. However, an unequivocal causal relationship cannot be established between Syrah decline and any of these viruses, although the possibility that certain virus or genetic variants, or both in combination, may contribute to the disease cannot be excluded. Gleaning all information that is available to date, we feel that the traditional approach and an insistence on finding a single cause for such a complex disorder in a woody perennial fruit crop involving grafting will prove to be futile. We hope that this study offers new conceptual perspectives on the etiology of this economically important but enigmatic disease complex that affects the global grape and wine industry.

## 1. Introduction

*Vitis vinifera* cv. Syrah, also known as Shiraz, is a dark-berried grape cultivar grown throughout the world, with concentrations in France, Australia, and the US. It was estimated in 2017 to be the world’s 7th most grown wine grape with around 142,450 hectares worldwide and continues to rise in popularity in recent years [1]. However, worldwide Syrah production has been negatively affected in recent decades by the occurrence and spread of Syrah decline.

Syrah decline was first identified in the cultivar Syrah in the 1990s in Southern France [2] and has now been reported in multiple countries, including the USA [3], Australia [4], South Africa [5], Spain [6], and Canada [7]. The economic impact due to this disease has not been documented but is believed to be high due to its wide distribution, severe damage to vine health, reduction in yield and quality of grapes, as well as death of vines. The progression of Syrah decline begins with swelling at the graft union, cracking/grooving of the woody cylinder, stem necrosis, followed by reddening of the canopy, failure in fruit ripening, and the death of the affected vine within 4 to 10 years [3,8]. In general, crevasses/cracking are not always sufficient to cause vine death, which typically occurs when vines develop both cracks/crevasses and red canopies [9].

Though first identified three decades ago, the cause of this disease remains elusive. A research group in France has systematically investigated many possible causes, including grafting methods, rootstocks used, hormone concentration, and bacterial/fungal/viroidal infections [2,8,10,11]; yet none of these could be definitively linked to Syrah decline. Many viruses have been detected in decline-affected vines, yet no clear and definitive association between any of the viruses and Syrah decline could be established to date. Interestingly, a genetic variant of Grapevine rupestris stem pitting-associated virus (GRSPaV), SA2, was initially reported to be associated with Syrah decline (aka Shiraz decline) in South Africa [5]. However, this claim was modified in a later publication stating that Syrah decline is unlikely to involve any of these viruses [12]. Grapevine Syrah virus-1 (GSyV-1), then a newly identified virus, was discovered through high-throughput sequencing (HTS) of Syrah grapevines with decline symptoms [13] (Al Rwahnih et al., 2009). However, a subsequent investigation concluded that GSyV-1 was unlikely the causal agent of Syrah decline [9]. 

GRSPaV is the most common virus in grapevines and the more likely candidate as a causal agent. A strain of GRSPaV, SY, was detected in declining Syrah vines in the USA [14] and Australia [4], implicating its possible involvement. Similarly, another strain of GRSPaV, strain PN, was detected in declining Pinot noir vines in California [15], indicating the possibility that different GRSPaV strains might be involved in the disease. In a comprehensive study, Beuve et al. (2013) [9] attempted to find possible correlation between different genetic variants of GRSPaV and Syrah decline through cloning and subsequent sequencing of RT-PCR products amplified from Syrah vines exhibiting different levels of sensitivity to Syrah decline. Though variants of the SY lineage was predominant in vines of the highly sensitive group, statistical analysis did not reveal significant relationship between any GRSPaV variants and vine sensitivity to the disease. 

In this study, we attempted to investigate the recent outbreak of Syrah decline observed in commercial vineyards in the Niagara region (Ontario, Canada) in a hope to probe into the possible involvement of known viruses to Syrah decline. We found that variant SK704-A of GRSPaV was detected in a majority of vines showing decline. Though we cannot conclude that any of the viruses we detected is the sole cause of the disease, it cannot be excluded that certain strains of GRSPaV may play a role in the development of Syrah decline. We propose a theory that brings together the practice of grafting, viral infection, water deficiency, blockage to sugar translocation and the related anthocyanin accumulation to account for the underlying mechanisms for the disease. 

## 2. Materials and Methods

### 2.1. Plant Materials

At the request of a winery in Niagara, Southern Ontario, Canada where Syrah is a signature wine product, we launched an investigation on the possible cause of Syrah decline that was observed by this winery among two vineyard blocks where Syrah was grown. Disease surveys were conducted in June and September of 2014 and 2015 in two Syrah vineyard blocks, both of which were established in 2001. Vineyard block 1 had Syrah clone 1 grafted onto rootstock 3309 Couderc (3309C) while vineyard block 2 had Syrah clone 100 also grafted onto the same rootstock. A follow-up survey was conducted for block 2 in September 2017. Leaves and canes were collected from both symptomatic (showing red canopy) and asymptomatic (without red canopy) vines from each block for virus detection. Additional samples were collected from two other vineyard blocks of Syrah clone 100 and used in the detection of grapevine asteroid mosaic-associated virus (GAMaV) and grapevine rupestris vein feathering virus (GRVFV).

### 2.2. Isolation of Total Nucleic Acids

Total nucleic acids were isolated from cane phloem tissues using the Spectrum Plant Total RNA Kit (Sigma) with minor modifications, as described in [16,17]. The purity and concentration of the total nucleic acid preparations were assessed with a NanoDrop spectrophotometer (ND-1000, Thermo Scientific, Waltham, MA, USA). The nucleic acids used for HTS was also evaluated with Bioanalyzer (Agilent) microfluidic electrophoresis chips. 

### 2.3. PCR and RT-PCR

Total nucleic acids were used as template for PCR or RT-PCR to test for 14 viruses that were shown to be present in commercial wine grapes based on a survey we conducted earlier [16]. Broad-spectrum primers were designed to target all strains for each virus for which the genome sequences were available and used in Xiao et al. (2018) [16] (Table 1). Strain-specific primers were designed for this study and also used to amplify two strains each of GRSPaV (SY and PN) and GRBV (NY-147 and NY-701) (Table 1). In addition, a new primer pair was designed for GVB and used in this study (Table 1). PCR and RT-PCR were performed according to the procedures described previously [17]. As a result of the small genome size, the genome of grapevine red blotch virus (GRBV), though a single-stranded circular DNA, would be included in the RNA preps that were isolated using the total RNA isolation kit from Sigma. Furthermore, as an essential stage of DNA virus replication, GRBV will produce mRNAs for the translation of viral proteins. 

### 2.4. High-Throughput Sequencing

Total nucleic acids were isolated from sample 93-21 (with red canopy) and 93-26 (without red canopy). The nucleic acid preparations were sent to the Donnelly Sequencing Centre of the University of Toronto (Ontario, Canada) for NGS. Total nucleic acids were subjected to the removal of ribosomal (r)RNA using Ribo-Zero rRNA Depletion Kit (Illumina). The rRNA-depleted nucleic acid samples were used for the construction of cDNA libraries using the TruSeq RNA Library Prep Kit from Illumina. An Illumina HiSeq 2500s sequencer was used to generate 51-bp paired-end reads. Both sets of reads were analyzed in iPlant Discovery Environment [18] (Goff et al., 2011). Reads were first mapped to the grapevine reference genome (*V. vinifera*, PRJEA18785) using TOPHAT2-SE of iPlant. Non-grapevine sequence reads were mapped to reference virus genome sequences using the BLAT tool of iPlant. These sequence reads were also de novo assembled into contigs in CLC Genomic Workbench (Qiagen). The BLAST tool was used to compare the contigs with complete sequences of viruses and viroids. To obtain complete or partial genome sequences of the viruses and variants that were identified, the de novo assembled viral contigs were compared with known viral genome sequences and assembled manually.

### 2.5. Genetic Diversity Analysis of GRSPaV

A part of the genome region encoding the RNA-dependent RNA polymerase (RdRp) domain of GRSPaV was amplified from five vines with red canopy (93-21, 93-27, 93-32, 210-3, and 210-5) and six vines without red canopy (93-22, 93-26, 93-28, 93-30, 93-31, and 210-4) using primers RSP35 & RSP36 and cloned into pGEM-T Easy (Promega) as described in Terlizzi et al. (2011) [19]. Eight or more clones from each vine were sequenced at the Genomics Facility, University of Guelph. Corresponding sequences were retrieved from GenBank for each of the following GRSPaV variants: GG (JQ922417), LSL (KR054735), VF (KT948710), SK704-A (KX274274), PN (AY368172), BS (AY881627), SG1 (AY881626), SY (AY368590), MG (FR691076), JF (KR054734), and WA (KC427107) and included in the phylogenetic analysis. Sequences were aligned using ClustalW with MEGA 7 software. Both Maximum Likelihood and Neighbour Joining methods were performed with 1000 bootstrapping replicates for phylogenetic analysis [20].

## 3. Results

### 3.1. Syrah Decline in Ontario Vineyards

Over a five-year period prior to the commencement of this study, the vineyard owner observed that the Syrah vines from two vineyard blocks declined in vigor, that berry yield and quality decreased year after year, and that some vines succumbed to freezing temperatures over the winters. To investigate the cause(s), both vineyard blocks were surveyed for the first time in June 2014; 95% of vines had yellowish leaves. In September of 2015, we conducted a follow-up survey of the two vineyard blocks and observed that 95% of vines had leaves exhibiting different levels of red discolouration with 25% of the vines having the entire canopies turning red (Figure 1A). Survey results for vineyard block 2 over a three-year period are shown in Table 2. Vines with red canopy had their entire leaf blades (including veins) turning uniformly red, bore fewer or no fruits, and developed necrosis in some of the canes (Figure 1C). Examination of the graft union revealed swelling, cracking, and pitting of woody tissue (Figure 1E). The disease severely affected vine vigour as well as the yield and quality of fruits. 

### 3.2. Several Viruses Were Detected in Declining Syrah Vines through RT-PCR/PCR

Initially, PCR and RT-PCR were used to test for viruses present in vines exhibiting a red canopy, one of the hallmarks of Syrah decline. Fourteen viruses were targeted in the RT-PCR tests. Total nucleic acids were isolated from cambial scrapings of 10 vines each for clone 1 and clone 100, all presenting different degrees of red discolouration (Table 3). Isolated nucleic acids were used as template for RT-PCR tests. As described earlier, the small genomic DNA of GRBV was also present in these total nucleic acid extracts and can be detected directly with PCR. As shown in Table 3, GRSPaV and grapevine leafroll-associated virus 3 (GLRaV-3) were detected in all 20 samples tested. GRBV was detected in all 10 samples from clone 100 and 6 of the 10 samples from clone 1. Grapevine Syrah virus 1 (GSyV-1) and grapevine Pinot gris virus (GPGV) were detected in 1 and 3 of the 20 samples, respectively (Table 3).

RT-PCR with strain-specific primers was used to detect SY and PN variants of GRSPaV in these 20 samples. While variant PN was not detected in any of the samples, variant SY was detected in all 20 samples, including those without red canopies (Table 3). No correlation could be definitively established between variant SY and Syrah decline observed in our study. However, we cannot establish a correlation between GRSPaV-SY titer and Syrah decline. 

### 3.3. Detection of Viruses in Declining Syrah Vines through HTS Analysis

Based on the results described above, we were unable to conclude that any of the viruses targeted by PCR/RT-PCR was responsible for the Syrah decline in Ontario. To further explore the possibility that Syrah decline might be due to uncharacterized virus(es) and/or viroid(s), we conducted HTS analysis. Sample 93-21, a vine of Syrah clone 100 with red canopy, and 93-26, a vine of the same Syrah clone but without red canopy, were subjected to RNA-Seq analysis. A total of 5.7 × 10^7^ and 6.9 × 10^7^ sequence reads were obtained from ribo-depleted cDNA libraries derived from 93-21 and 93-26 samples, respectively (Table 4). Data analysis revealed that a total of 5.8 × 10^6^ reads from 93-21 and 7.4 × 10^6^ reads from 93-26 did not map to the grapevine reference genome (Table 4), and hence were expected to contain sequences derived from pathogens including viruses and viroids. These non-grapevine reads were then mapped to the reference genomes of viruses and viroids available in GenBank. A total of 30,506 reads from sample 93-21 and 60,260 reads from sample 93-26 were identified as virus/viroid sequences. Both samples had reads that matched GRSPaV, GRBV, and GLRaV-3, in agreement with results from RT-PCR and PCR. Also as expected, grapevine yellow speckle viroid 1 (GYSVd1), GYSVd2, and hop stunt viroid (HSVd) were detected in both samples. What was not expected though was the identification of two new viruses: GAMaV was present in both samples, whereas GRVFV was detected only in 93-21 and not 93-26 (Figure 2). Another difference between the two samples is that the percentage of GRBV-associated reads in 93-21 is only a sixth of that in 93-26 (Table 4).

Sequence reads not mapping to the grapevine reference genome were also subjected to de novo assembly. The resulting contigs were BLAST-searched against the available reference genomes of viruses and viroids using CLC Genomic Workbench. Both 93-21 and 93-26 had contigs matching GRSPaV, GRBV, GLRaV-3 and GAMaV, as well as three viroids (GYSVd1, GYSVd2 and HSVd), while only 93-21 had contigs matching GRVFV, as expected (Table 5). Sequence contigs were also compared with the known genome sequences of these viruses and their genetic variants and assembled into draft genomes. As shown in Table 5, five GRSPaV variants were found in both samples, with -SY being a minor constituent in both samples, as judged by the low sequence coverage (17% for 93-21; 12% for 93-26 (Table 5). It is important to point out that very low genome coverage was detected for the contigs related to SY. This raises questions as to which variants of SY were indeed present in these two samples. Complete genome coverage was obtained for GRBV for both samples, with isolate NY-147 identified in vine 93-21 and isolate NY-701 in vine 93-26 (Table 5). These two GRVB variants differ by 8% in their genome sequences (data not shown).

### 3.4. GAMaV and GRFVF in Declining Syrah Vines

To test the possibility that either GAMaV or GRVFV was responsible for Syrah decline, a pair of primers was designed for each virus and used in RT-PCR to detect each virus in 22 vines with red canopy and 22 vines without red canopy. As shown in Table 6, GAMaV was detected in 82% of the vines with red canopy and 32% of those without red canopy, whereas GRVFV was detected in 23% of the vines with red canopy and 41% of those without red canopy. 

### 3.5. Analysis of GRSPaV Genetic Variants in Declining Syrah Vines

The population structure of GRSPaV variants in both types of vines was further examined by phylogenetic analysis of clones derived from RT-PCR using primers RSP35 & RSP36 (Table 1) [19] (Terlizzi et al., 2011) from five vines with red canopy and six without red canopy. A total of 95 clones (42 clones from vines exhibiting red canopy and 53 clones from vines without red canopy) were sequenced and resulting sequences were subjected to phylogenetic analysis. These resulting sequences were distributed among four phylogenetic clades: VF1, JF, GG/SG/BS, and SY (Figure 3). As shown in Figure 3, a large majority of the clones (62 out of the 95 clones) fall into two clusters, one represented by VF1 and the other by JF. Twenty four clones fall within the GG/SG/BS cluster, and seven clones fall within the SY cluster. Notably, 67% of clone sequences in clade VF1 (24 out of 36 clones) were from vines of red canopy whereas only 15% of clones (4 out of 26 clones) in clade JF were from vines with red canopy. The number of clones in clades GG/SG/BS and SY from declining vines is similar to that from non-declining vines (Figure 3). 

## 4. Discussion

Syrah decline has become a major concern to the grape/wine industry in many grape-growing countries [8,12,13,14]. A similar disease was recently reported in two Syrah clones in Ontario, Canada [7]. The primary symptoms involve damage to the trunk around graft union, including swelling, cracks, grooves, and necrosis; secondary symptoms include uniformly red discoloration of leaves and canopy starting in late summer or early fall and the eventual death of infected vine. It is important to note that red canopy represents late-stage Syrah decline, and the absence of a red canopy may mean that the disease is still in its early stages. The cracking and stem damage observed in Syrah decline resembles, to a certain extent, symptoms of rugose wood (RW) complex and graft-incompatibility [12,21]. We propose that Syrah decline may be a unique manifestation of graft union abnormality specific to Syrah. Despite extensive efforts by several research groups since the recognition of the disease, the etiology of Syrah decline remains a mystery. 

In this study, we attempted to find possible links between viruses and Syrah decline observed in two Syrah clones in Niagara, (Ontario, Canada). We used RT-PCR, RT-qPCR and HTS to unravel all viruses and viroids present in declining Syrah vines. We detected multiple viruses in Syrah vines showing decline, which include GRSPaV, GLRaV-3, GRBV, GAMaV and GRVFV as well as three viroids. 

GLRaV-3 was detected in all vines assayed regardless of whether they displayed red canopy or not. However, no difference was observed either in the variants or in the titer of GLRaV-3 between vines with red canopy and those without. Two independent studies conducted in different countries on the virome of Syrah decline-affected vines did not reveal the presence of GLRaV-3 [9,13]. Based on the results from this study and those by others, we conclude that GLRaV-3 is unlikely responsible for Syrah decline, at least when acting alone. Similarly, GRBV, a DNA virus of the family *Geminiviridae* recently discovered [16,22,23], was also detected in most of the vines, regardless of having red canopy or not. Since most of the vines also exhibited symptoms of red blotch earlier in the season when red canopy did not yet develop, it was expected that these vines would test positive for GRBV. Furthermore, the titer of GRBV in vines with red canopy was lower than in those without red canopy, lending further evidence that GRBV is not likely the cause of Syrah decline. 

HTS analysis revealed two other viruses, GRVFV and GAMaV, in vines exhibiting Syrah decline. To rule out the possibility that GRVFV and/or GAMaV may be involved in Syrah decline, we conducted a preliminary survey for their presence in Syrah vines. GRVFV was detected in only 23% of the vines with red canopy and 41% of the vines without red canopy, suggesting that GRVFV is unlikely the cause of the disease. GAMaV was detected in 82% of vines with red canopy and only 32% of those without red canopy. It is of interest to note that GAMaV was also detected in California through HTS in vines with Syrah decline [13]. However, read counts of GAMaV were very low in both HTS studies, raising legitimate doubts about its involvement as the causal agent of Syrah decline. 

In line with research conducted earlier by different research groups, GRSPaV was detected in all vines assayed, including those with red canopy and those without red canopy. Genetic diversity analysis and HTS revealed the presence of five variant groups of GRSPaV (Figure 3). Interestingly, variants related to isolate SK704-A appear to have a greater presence in declining vines compared to those without red canopy: 24 clones from vines showing Syrah decline and 12 clones from vines without Syrah decline (Figure 3). However, definitive relationship between any genetic variant group of GRSPaV and Syrah decline cannot be established with confidence. GRSPaV has been persistently detected in vines affected by Syrah decline, although the causative role GRSPaV in Syrah decline has never been established. In a pioneer work, Lima et al. (2006) [14] investigated the cause of a decline in a field selection of Syrah in California. This affected vine exhibited typical symptoms of Syrah decline (weak growth, red canopy and enlarged trunk above the graft union) but was negative for all 15 viruses tested, except a highly divergent variant of GRSPaV, which was designated SY. To test the possibility that some yet unknown viruses might be involved in the disease, a follow-up study involving HTS was conducted by Al Rwahnih et al., 2009 [13]. GRSPaV, along with three other viruses [GSyV-1, GRVFV and GLRaV-9 (now a strain of GLRaV-4) and three viroids were detected. Goszczynski (2010) [12] examined the distribution of genetic variants of GRSPaV and two viruses of the genus *Vitivirus* (GVA and GVB) in several Syrah clones as well as three commonly used rootstocks (Mgt 101-14, Richter 99, Richter 100) using broad-spectrum primers targeting members of the genera *Foveavirus* and *Vitivirus*. Five groups of genetic variants of GRSPaV were detected, one of which (designated SA2) was closely related to strain SY. In another study, Beuve et al. (2013) [9] attempted to establish a correlation between GRSPaV strains and the severity of Syrah decline symptoms among a collection of Syrah clones through RT-PCR, cloning, sequencing and statistical analysis. They showed that viral variants of the SY lineage were predominant in the high sensitivity group of Syrah clones. Surprisingly and without explanation, they concluded that none of the GRSPaV strains could be correlated to Syrah decline [9]. Based on findings from our present study, a genetic variant group possibly contributing to Syrah decline may be SK704A/VF1 (Figure 3). Certainly, this supposition needs to be tested. Other viruses may possibly co-infect vines alongside GRSPaV to worsen decline. Availability of infectious cDNA clones corresponding to this and other GRSPaV strains as well as experimental systems with which to launch grapevine infection with infectious clones will be of critical importance to such studies.

The fact that Syrah decline seems to primarily affect Syrah over other cultivars implies that Syrah may be particularly susceptible to the disease due to some genetic factors and water status in the soil. Evidence in support of this comes from observation of Syrah decline symptoms in Syrah grapevines tested free of all major viruses and viroids and that were grown under conditions where water stress was purportedly absent [24] (Puckett et al., 2018), though no further information is available in the literature to validate this claim. Compared to other red wine grape cultivars, Syrah is particularly susceptible to water stress because of its low efficiency in water use. Syrah is classified as anisohydric, i.e., having poor control of water loss through stomata [25,26,27]. Elevated temperatures, typically accompanied by drought [28], cause Syrah to increase the openings of their stomata instead of closing them [29], leading to even greater water loss. Syrah decline tends to affect grapevines in regions with low water content in the soil and has been noted to occur more frequently in shallow soils [8,30]. Southern France, where Syrah decline was first observed, has been experiencing a significant negative trend in soil moisture over time, which could explain the increasing incidence of Syrah decline [31] (European Environment Agency 2016). Satellite measurements of soil moisture indicate that regions in North America where Syrah decline has been reported, e.g., California and the Niagara Peninsula, often exhibit significantly lower water content in the soil during summer and early fall as compared to other regions [32] (Agriculture and Agri-Food Canada 2016). The data reported in this study was collected during late summer/early fall; water stress could have contributed to the symptom development.

While Syrah-specific physiology and drought likely render Syrah more prone to the disease, it is possible that infectious agents such as viruses may trigger or exacerbate Syrah decline (Figure 4). Infections with certain viruses or strains of a given virus, especially mixed infections with multiple viruses, may contribute to the development and acceleration of Syrah decline. It is worth noting that different genetic variants of GRSPaV were detected in the samples assayed here. This is in line with several previous reports on mixed infections of grapevines with distinct viral variants of GRSPaV [19,33,34,35,36]. Damage caused by viral infections at the graft union, as observed in the RW disease complex, would obstruct the transport of sugar via the phloem and water through the xylem, leading to the blockage of sugar export out of leaves and water deficiency in the canopy [37,38]. Several viruses of the family *Betaflexiviridae* have been implicated in various diseases of the RW complex [39,40,41]. It is logical to suggest that Syrah decline is a unique and more extreme manifestation of the RW/graft-incompatibility syndrome that occurs in Syrah cultivar. As production of wine grapes would involve grafting of a scion cultivar onto a rootstock, the rootstock infected with certain viruses can serve as source of virus that would be transmitted to the scion upon grafting. If these viruses were highly pathogenic to the scion, damage to the tissues at the graft union would ensue. This in turn would result in the various symptoms characteristic of Syrah decline, such as swelling, pitting, grooving, necrosis and ultimately the reddening of leaves throughout the entire canopy. To this end, different GRSPaV variants have been detected in several commonly used rootstocks, such as Mgt 101-14, Richter 110, Riparia Gloire, among others [12,33] and our unpublished data). Therefore, the infection status of the rootstocks must be considered when studying the etiology of enigmatic diseases of woody fruit crops such as Syrah decline.

Here, we propose a theorem to account for the mechanisms underlying the development of Syrah decline (Figure 4). Syrah decline, as originally defined by French researchers [2], is an issue related to injury at the graft union between a scion and a rootstock that result in pits, grooves, cracks, and necrosis. Such an injury could be produced by pathogens including viruses. As a compensatory response, excessive cell division occurs at the graft union, leading to swollen union. Under conditions when grapevine experience sever water deficit due to low water content in the soil, and uncontrolled transpiration as is the case for Syrah grapes, transport of sugar out of leaves will also be impaired, resulting in the accumulation of glucose in leaves. In turn, this triggers the conversion of the colorless anthocyanidins to anthocyanins, manifesting in the form of red discoloration of the canopy. Supply of photo assimilates to shoot stems and roots would be impaired, limiting the development of roots and buds. Affected vines may die several years later as a result of the cumulative effects and worsening symptoms. 

Syrah decline, like decline symptoms observed in other grafted woody fruit crops, i.e., rapid apple decline [41,42,43] and rapid Prunus decline [44], is a complex disorder that may have a variable etiology. The direct cause is injury at the graft union together with water deficit in the plant. Continued insistence on, and further attempts to find, a single pathogen for Syrah decline will prove to be futile.

## 5. Conclusions

In closing, the paradigm that there must be a single cause for a disease does not work for many complex diseases of woody perennial plants such as Syrah Decline. Syrah decline is a complex disease to which Syrah clones are most susceptible due to multiple factors, including sensitivity to infection with certain virus(es) and low efficiency of water use. These factors cannot be considered in isolation when it comes to the etiology of the disease. Though viruses are unlikely the sole cause of Syrah decline, infection by certain viruses individually or in combinations, particularly mixed infections involving GRSPaV, may result in damage to the graft union. Such damage, when advanced beyond a certain threshold, would block sugar export from leaves, restricting carbohydrate supply to the roots causing root starvation, reduced water uptake from the soil, and enhanced anthocyanin synthesis in leaves. As a result, severely affected vines will exhibit red canopy later in the season and succumb to freezing temperatures over the harsh winter as frequently experienced in Ontario, Canada. 

Contemporary viticulture is unique compared to other plant crop systems in that grapevines are perennials and that it requires grafting between scions and rootstocks. The potential role of rootstocks for use in grafting must be considered as they could serve as covert carriers of certain viruses that may cause damage to the scion. Furthermore, mixed viral infections are common in grapevines [9,39,45] and synergistic effects among co-infecting viruses may worsen plant health. A new and exciting research area in grapevine virology is on plant-virus interactions at the level of mixed infections with multiple viruses and viral strains, which may offer insights into mechanisms underlying the growing problem faced by the global grape and wine industries, Syrah decline.

## Figures and Tables

**Figure 1 viruses-15-00023-f001:**
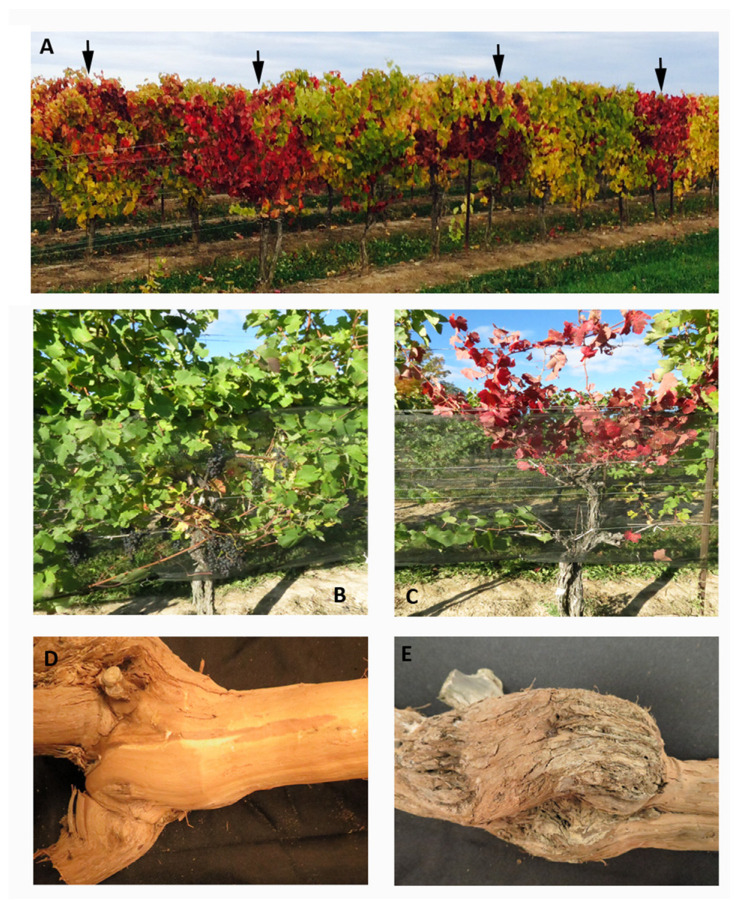
Symptoms of Syrah decline observed in a commercial Syrah vineyard in Niagara, Ontario, Canada. (**A**) An infected vineyard block with Syrah decline. Arrows indicate the vines with a red canopy. (**B**) A vine without a red canopy. (**C**) A vine with a red canopy. (**D**) The graft union of a vine without a red canopy. (**E**) The graft union of a vine with a red canopy. Typical symptoms of Syrah decline include a red canopy (**C**), swelling and crevasses at the graft union (**E**).

**Figure 2 viruses-15-00023-f002:**
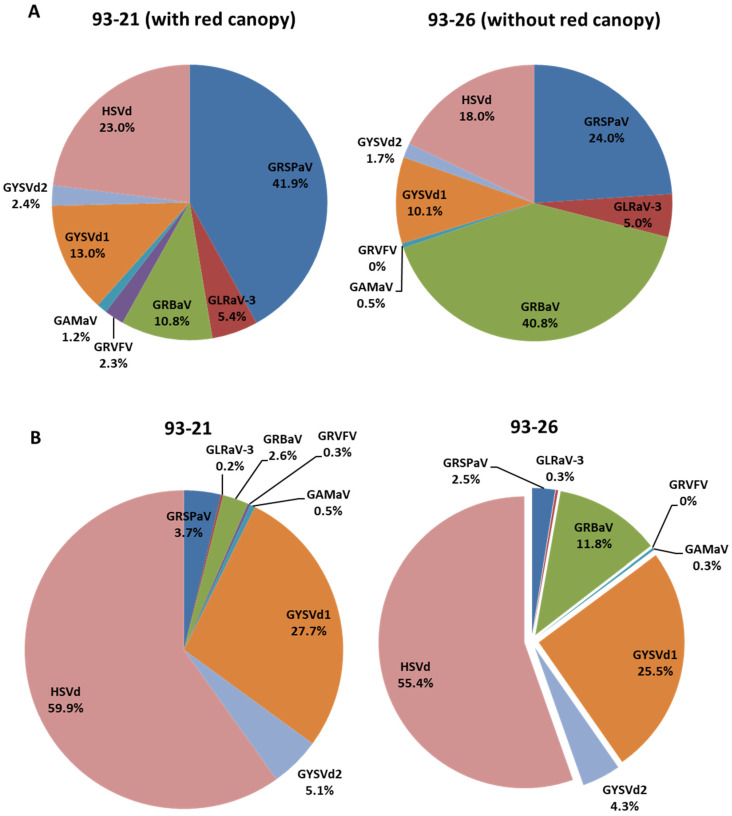
The distribution of viruses and viroids identified in 93-21 and 93-26 (both of Syrah clone 100) revealed by HTS. (**A**) The percentage of identified viruses and viroids was calculated as the number of sequence reads corresponding to each virus or viroid divided by the total virus and viroid reads in each sequencing library. (**B**) The percentage of identified viruses and viroids was calculated as the genome copy number of each virus or viroid over the total genome copy number of all viruses and viroids in each library. The genome copy number of each virus or viroid was calculated by multiplying the number of sequence reads specific to the virus by 51 (the number of nucleotides in each read), which was then divided by the size of the genome of the virus or viroid.

**Figure 3 viruses-15-00023-f003:**
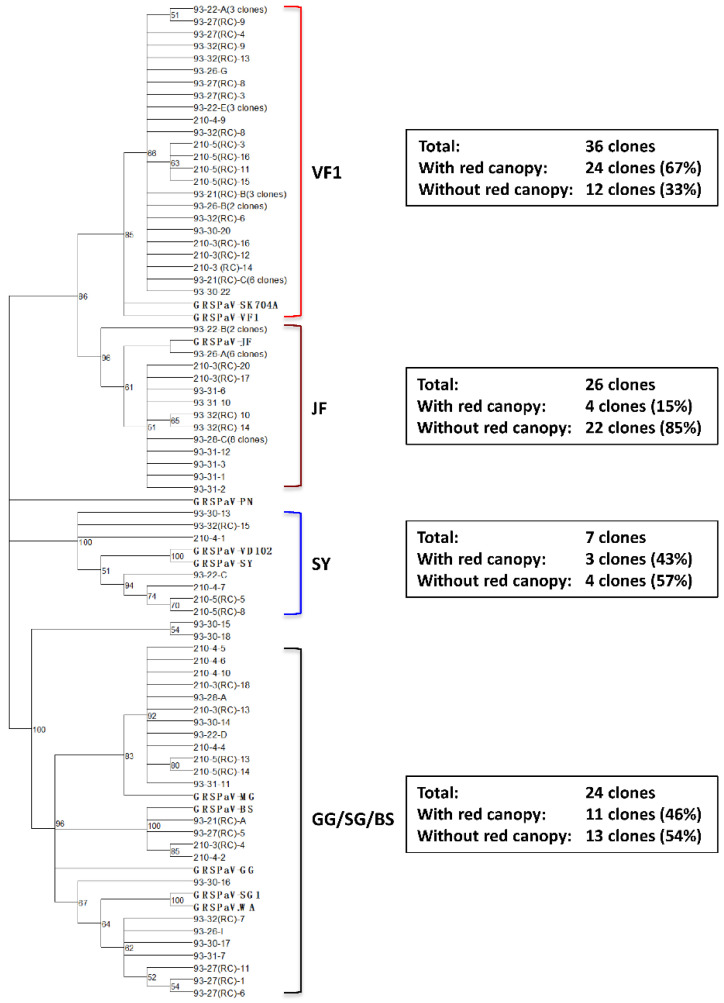
Distribution of clones derived from RT-PCR using primers RSP35 and RSP36 from vines with and without decline. A 476 bp region of GRSPaV RdRp was amplified in PCR using primers RSP35-RSP36 from five vines with red canopy (93-21, 93-27, 93-32, 210-3 and 210-5), which is indicated with RC in brackets, and six vines without red canopy (93-22, 93-26, 93-28, 93-30, 93-31 and 210-4). Phylogenetic analysis was conducted in MEGA X by using the maximum likelihood method with 1000 bootstrap replicates. The tree is drawn to scale, with branch lengths measured as the number of substitutions per site. Corresponding sequences from 11 reference GRSPaV isolates were retrieved from GenBank and included in the phylogenetic analysis. A summary of the numbers of clones from vines with decline (red canopy) and without decline (with no red canopy) in each clade is shown to the right of each clade.

**Figure 4 viruses-15-00023-f004:**
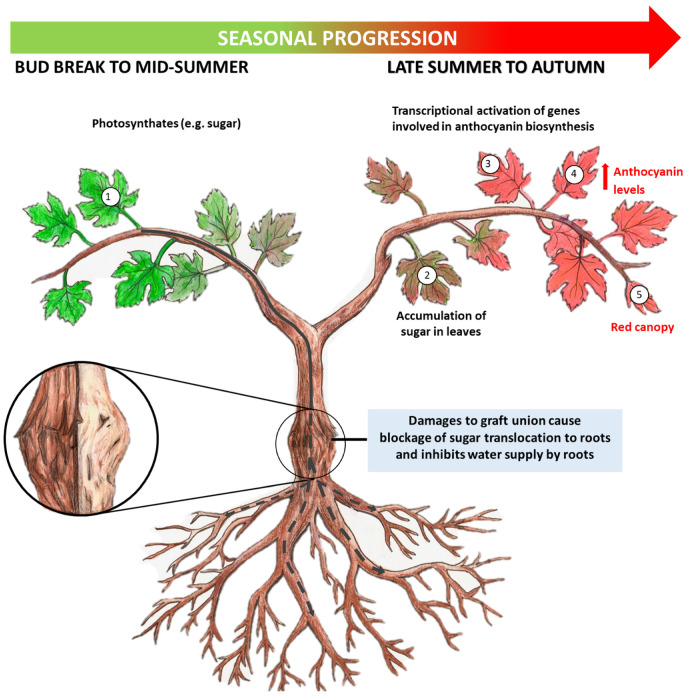
Proposed model of the pathogenesis of Syrah decline. Syrah decline is a physiological response to alterations at the graft union induced by viruses, which can be from the scion, the rootstock or both. Though it is a possibility that infection with a single virus may cause Syrah decline, it is more likely that mixed infections with multiple viruses and viral strains are involved. Viral infections induce cytopathic effects to cambium cells at the graft union, leading to disjunction and incompatibility between the scion and the rootstock. With time, this leads to the development of swelling, cracking, grooving and necrosis at the graft union. Consequently, sugar export from leaves will be impeded, leading to red discoloration of leaves on a portion or the entirety of the vine. In essence, Syrah decline is simply an extreme manifestation of rugose wood syndrome, which is unique to Syrah cultivars. For a detailed account of the model, please refer to Section 4.

**Table 1 viruses-15-00023-t001:** Primers used in PCR and RT-PCR for the detection of commonly targeted grapevine viruses. All primers used here were used in a previous viral survey (Xiao et al., 2018), with the exception of the following four primer pairs, which were newly designed in this study: GRSPV-SY, GRSPaV-PN, GRBV and GVB.

Target	Primer	Sequence	Size (bp)	Gene
GRSPaV	RSP35	AGRYTTAGRGTRGCTAARGC	477	Replicase
	RSP36	CACATRTCATCVCCYGCAAA		
GRSPaV-SY	SY1659F	TAAGATGGCCTTGGGTGTGG	469	Replicase
	SY2127R	ATTTATGGGATGGGCACATG		
GRSPaV-PN	PN1701F	CTTCTTGGTGAACAGCGCC	487	Replicase
	PN2187R	AACAAATTGCCTCACAAGCC		
GLRaV-1	LR1-580F	CCAGAYACNGARAGYAAAGAAG	201	CP
	LR1-780R	CTCGTTCGGCYTYAACTTTCC		
GLRaV-2	LR2-14568F	RCDATGGAGYTRATGTCYGA	525	CP
	LR2-15092R	AGCGTACATRCTYGCRAACA		
GLRaV-3	LR3CP107F	TCTTAAARTAYGTTAAGGACGG	301	CP
	LR3-CP407R	GGCTCGTTAATAACTTTCGG		
GLRaV-4	LR4-13269F	GGACAATTTAGGTAATGTWGTRGCTAC	490	P23
	LR4-13758R	TATCCTCAGWGAGGAARCGG		
GLRaV-7	LR7-12163F	CTAGTGAATTACACCGAGAAGTC	550	CP
	LR7-12712R	GTGACTTGGCACGCATGTATC		
GRBV	GRBV1097F	ACGAGGAATCGTTTGAATCG	235	CP
	GRBV1331R	TAAACGTATGTCCACTTGCAG		
GRBV-147	RB147F2422	AGGTTTCGTTGTGCTGAGCT	303	C1
	RB147R2724	CGTCGAGCACGGTACAAAG		
GRBV-701	RB701F133	GGTGGCCGAATATATGCTTAA	505	V2
	RB701R638	CAACGCGTCTAGTCAGTTGA		
GSyV-1	GSyV5725F	ATGATGCAACCGACCCTTCC	671	CP
	GSyV6395R	TGGAGGCTTTATTCAGAGAG		
GPGV	GPGV6586F	GAYATGTCGATTCGTCAGGAG	436	CP
	GPGV7021R	CAACGCGTCTAGTCAGTTGA		
GVA	GVA6538F	TCTTCGGGTACATCGCCTTG	325	CP
	GVA6862R	TCRAACATAACCTGTGGYTC		
GVB	GVB6411F	GTGTAYGARACAATAAGCAAGC	721	CP
	GVB7131R	TAGCCCTYCGTTTAGCCGCA		
GFLV	GFLV3135F	TTGAGATTGGWTCYCGTTTC	558	CP
	GFLV3692R	CTGTCGCCACTAAAAGCATG		
ArMV	ArMV2291F	CRGGTATTACGTGGGTTATGAG	340	CP
	ArMV2582R	GCTGCCTCAAACTCAGCATA		
GRVFV	GRVFV5646F	GTYGAARTCTCTCTCTTCTCCC	389	Replicase
	GRVFV6034R	ATTATGAGAGCAACCCACTGGAAG		
GAMaV	GAMaV6165F	CTCGCGCTCCTCGCATTGTT	467	Replicase
	GAMaV6631R	CGTGACGAGGTTGGTCCCA		

**Table 2 viruses-15-00023-t002:** Syrah decline frequency in Ontarian vineyards. The survey was conducted in the month of September in 2014, 2015 and 2017 in vineyard block 2, where Syrah clone 100 grafted on the rootstock 3309C was planted. NA: not available.

Year	Total No. of Vines Surveyed	No. of Vines with Red Canopy	No. of Dead Vines	No. of Replanted Vines
2014	510	127	NA	NA
2015	209	29	47	133
2017	229	5	31	193

**Table 3 viruses-15-00023-t003:** RT-PCR and RT-qPCR detection of viruses from Syrah vines with and without Syrah decline symptoms. All vines tested also had symptoms of infection with GRBV.

Sample	Clone	Symptoms	GPGV	GSyV	GRSPaV		GLRaV-3	GRBV
					All Strains	SY		
93-1	Clone 1	Red canopy	-	-	+	+	+	-
93-2		Red canopy	-	-	+	+	+	-
93-3		Red canopy	-	-	+	+	+	+
93-4		Red canopy	-	-	+	+	+	-
93-5		Red canopy	-	-	+	+	+	+
93-6		Red canopy	-	-	+	+	+	+
93-7		Red canopy	-	-	+	+	+	-
93-8		-	-	-	+	+	+	+
93-9		-	+	-	+	+	+	+
93-10		-	+	-	+	+	+	+
93-21	Clone 100	Red canopy	-	-	+	+	+	+
93-27		Red canopy	-	-	+	+	+	+
93-32		Red canopy	+	-	+	+	+	+
93-22		-	-	-	+	+	+	+
93-25		-	-	-	+	+	+	+
93-26		-	-	-	+	+	+	+
93-28		-	-	-	+	+	+	+
93-29		-	-	-	+	+	+	+
93-30		-	-	+	+	+	+	+
93-31		-	-	-	+	+	+	+

**Table 4 viruses-15-00023-t004:** Summary of HTS analysis of the virome of two Syrah vines that either exhibited a red canopy (93-21) or were without a red canopy (93-26) and their corresponding viruses/viroids. Grouping and abundance of sequence reads obtained from both samples were determined by mapping the sequence reads to the virus/viroid database using BLAT. Note: sample 93-26 had more sequence reads for all viruses and viroids that were detected, with the exception of GRVFV, which was only present in sample 93-21.

	No. of Reads		% of Viral Reads	
	93-21	93-26	93-21	93-26
Reads mapping to *V. vinifera* genome	51,118,440	61,175,450	89.86	89.26
Reads not mapping to *V. vinifera* genome	5,767,366	7,363,547	10.14	10.74
GRSPaV	12,795	14,432	0.022	0.021
GRBV GLRaV-3	3285 1649	24,557 3033	0.006 0.003	0.036 0.004
GAMaV GRVFV	365 701	319 0	0.001 0.001	0.004 ―
GYSVd1	3968	6087	0.007	0.009
GYSVd2	724	1013	0.001	0.001
HSVd	7019	10,819	0.012	0.016
Total # of reads	56,885,806	68,538,997	―	―
Total # of reads related to viruses and viroids	30,506	60,260	0.054	0.088

**Table 5 viruses-15-00023-t005:** Summary of contigs assembled from sequence reads derived from two samples of Syrah clone 100 vine (93-21 with red canopy; 93-26 without red canopy) and their corresponding viruses and viral variants. Viral contigs obtained based on de novo assembly from each of the two samples were manually compared with known viral genome sequences and assembled into variant sequences. Data pertaining to viroids are not included here as they are commonly detected as “background” and are not known to cause major diseases in grapevine.

Virus	Variant/Accession	93-21 (With Red Canopy)			93-26 (Without Red Canopy)		
		Number of Contigs	Nucleotide Identity	Genome Coverage	Number of Contigs	Nucleotide Identity	Genome Coverage
GRSPaV	MG/FR691076	2	98	90.0	7	94–98	36
	BS/1Y881627	1	94	88.4	1	98	38
	JF/KR054734	8	98	65	4	97–98	43
	SK704A/KX274274	10	97–99	85	7	98–99	89
	SY/AY368599	3	95	94–97	3	17	12
GRBV	NY-701	-	-	-	1	99	100
	NY-149	4	99	100	-	-	-
GLRaV-3	623/GQ352632	5	99	14	7	99	23
GAMaV	USA9/AJ249357	3	92–93	61	3	93–94	62
GRVFV	NA/AF706994	4	69–84	88	-	-	-
		5	80–88	73	-	-	-

**Table 6 viruses-15-00023-t006:** RT-PCR detection of GRFVF and GAMaV from Syrah vines with and without Syrah decline symptoms.

Vines with Red Canopy	GAMaV	GRVFV	Vines without Red Canopy	GAMaV	GRVFV
93-1	+	+	93-8	-	-
93-2	+	-	93-9	-	+
93-3	-	-	93-10	+	-
93-4	+	-	93-22	-	+
93-5	+	-	93-25	-	-
93-6	+	-	93-26	+	-
93-7	-	-	93-28	+	-
93-21	+	+	93-29	+	-
93-27	-	-	93-30	-	+
93-32	+	-	93-31	-	+
210-3	+	+	210-4	+	-
210-5	+	-	210-6	-	+
ON427	+	-	ON417	-	-
ON428	+	+	ON418	-	-
ON429	+	-	ON419	-	+
ON430	+	-	ON420	-	+
ON431	+	-	ON421	-	-
ON432	+	-	ON422	-	+
ON433	+	-	ON423	-	-
ON434	-	-	ON424	-	+
ON435	+	-	ON425	+	-
ON436	+	+	ON426	+	-
# of positive samples	18	5	# of positive samples	7	9
Total # of samples tested	22	22	Total # of samples tested	22	22
Percentage of positives	82%	23%	Percentage of positives	32%	41%

## Data Availability

Sequences derived from HTS and genetic diversity analysis have not been deposited in public sequence database but will be deposited in GenBank immediately upon acceptance of the manuscript for publication.

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
