# Peer review of "Concerning the Etiology of Syrah Decline: A Fresh Perspective on an Old and Complex Issue Facing the Global Grape and Wine Industry"

_viruses, 2022, doi:10.3390/v15010023_

Round 1
Reviewer 1 Report
The manuscript of Xiao et al. describes the research focused on the identification of causal agent(s) of Syrah decline. As expected from the previous works, an involvement of a single viral/viroid pathogen responsible for the disease could not be established. Despite this fact, the results can be of interest for grapevine virologists. However, the manuscript is sometimes difficult to follow and should benefit from redrafting.
Following points need attention:
Line 98: please, describe the symptoms observed (especially, if particular). Or is it the “red canopy”?
Line 116: “would be” does not sound too scientific. Is this statement supported by a reference? Or has GRBV an RNA stage during its replication cycle in the plant cell?
Line 137: although the RSP235/RSP36 primer pair is degenerated, due to a huge diversity of GRSPaV, a possibility exist that these primers are not enough polyvalent to detect all virus variants. Is it possible to perform an in sillico analysis using available complete GRSPaV genomes (cca 300 genomes) to know, if there is a possible bias in detection
Line 150: it is quite unusual that the sequences are not deposited to the public database prior to manuscript submission
Line 153 and Table 1: please, redraft clearly that the reference to each primer pair is Xiao et al. 2018
Line 203: “Viral titers of GRSPaV variant SY in the samples varied considerably as judged by the intensity of the amplification products on the gel (data not shown).” I suggest to remove this sentence. In my opinion, the viral titers are cannot be linked to the intensity of PCR bands at all (no quantitative PCR was performed), The intensity of PCR band could be influenced for example, by imperfect annealing of primers due to the mismatches in the sequence targeted by primer etc., and not only by amount of PCR template.
Line 277: as shown in previous works (e.g. doi: 10.5423/PPJ.OA.07.2016.0158 and doi: 10.1007/s00705-018-3945-0 ) authors detected the mixed infection of different GRSPaV variants in the same plant), however, this is not later discussed, except a short opinion done in line 473.
Line 290: columns 6 and 7. It is hardly to understand to what the nucleotide identities refer. Can you explain it? Another point to discuss: some GRSPaV contigs have a low coverage to reference genomes (only 12% for SY in case of 93-26). Are authors sure about their analysis based on such low coverage (affiliation of contigs to a particular variant) as it was shown that recombination was found to shape the GRSPaV diversity (e.g. doi: 10.5423/PPJ.OA.07.2016.0158 ) and consequently, this could hamper the proper analysis? Can this point be discussed at least?
Line 322: “five phylogenetic clades: SK704-A, JF, MG, BS, and SY” where these clades defined previously (reference in this case), or this naming is used for the purpose of this work?
Author Response
Dear reviewer:
I would like to take this opportunity to thank you for the speedy and thorough review of our manuscript and for the large amounts of comments and suggested changes. I have carefully considered every comment and suggestion and incorporated all changes as suggested by the reviewers. In addition to the changes made based on the review comments, I took the liberty in going through the manuscript the last time and made additional changes to the manuscript. Most of these changes are technical and minor in nature, with the intention to achieve better clarity, accuracy, and information flow. All these changes are highlighted by using the Track Changes function.
Below I provide a detailed item-by-item response to each of the suggested changes you raised during the first round of review.
I would also like to note that GenBank accessions for complete or partial genome sequences for viruses identified through RNA-Seq are provided in the manuscript. We also reformatted both the in-text citation and the reference list according to the format required for viruses.
I very much appreciate your help with the review of our manuscript and the detailed and valuable suggestions. Please let us know if any further change or assistance is required. I shall do my very best to assist you in reaching your final decision regarding our manuscript.
Reviewer 2 Report
To the editor and authors:
Attempts have been made to find the association of a virus in Syrah Decline (SyD), but like previous papers it was in vain as they gain no no clear-cut answer. Overall, poor decisions were made on the selection of right controls. No control without GLRaV-3 was included. In one attempt, 24 symptomatic vines showed an association with the SK407-A strain of grapevine rupestris stem pitting association virus, while 12 asymptomatic vines also had the same strain of the virus. They linked this strain as a causative candidate.
Besides, since only one plant from each symptomatic and non-symptomatic vines was subjected to HTS, the deduced data may not be very conclusive. At least two plants from each group should have been used to gain conclusive evidence. However, since this study was done under the freezing climate condition of Ontario, which is unique for the Syrah habitat, the paper is worth considering provided the following are met:
1. Line 35, convert to hectares
2. Lines 49-50, the sentence is not clear.
3. Lines 60-62: Remove the following sentence and the reference all together:
4. ……potentially Apple stem pitting virus (ASPV) were initially reported to be associated with Syrah decline (aka Shiraz decline) in South Africa (Goszczynski 2007).
5. Line 85, delete “perhaps even a significant one”
6. Lines 99-102: reasons for that study and its relevance were not given, nor if the vines were showing SyD symptoms.
7. Line 104: You must write “total nucleic acids” rather than total RNA (see also lines 117-118).
8. Line 121: total nucleic acids.
9. Line 175: double full stop.
10. Line 180: double “with”.
11. Line 183: make it clear if NA means zero (no replanted vine).
12. Line 189: “…. decline. Fourteen.. “
13. Line 190: which season the samples were collected?
14. Line 191: you didn’t include a negative symptom control. Why? Syrah infected with GLRaV-3, always shows symptoms later in summer.
15. Line 204: Not valid, as you didn’t use an internal host plant marker RNA in duplex RT-PCR.
16. Line 219: Since these vines are infected with GLRaV-3 as well, they would show red canopy anyway. You must distinguish between red leaf symptoms induced by GLRaV-3 and those from the SyD infection. No vine negative for GLRaV-3, but positive for SyD has been included in this study.
17. Line 255: Figure 2 of pie shapes. Here, GRBaV should be GRBV.
18. Line 280: modify as “… was obtained for GRBV in both samples”
19. Line 284: “with red canopy”
20. Line 288: as ‘background viroids”
21. Line 419: this 24:12 cannot provide you with any convincing evidence about SyD. In fact, 12 as a negative number is too many compared to 24 as positive.
22. Line 421: Delete “only” , because 12 is 50% of 24.
23. Lines 442-444: no this is not convincing; it should be deleted.
24. Line 466: “America”
25. Line 494: Too many explanations in the Figure 4 caption. Some of them could be moved to “Discussion”. Numbers on leaves and black dashed lines on the tree were not depicted.
26. Line 495: you wrote …. induced by pathogenic viruses…. How do you know? We can’t see any evidence here.
Author Response
Dear reviewer:
I would like to sincere gratitude to you for the thorough and speedy review of our manuscript. We find the comments and suggested changes very helpful for us to improve the quality of our manuscript. I have carefully considered every comment and suggestion and incorporated all changes you have raised in the first round of review. In addition to the changes made based on the review comments, I took the liberty in carefully combing through the manuscript in its entirety and made additional changes to the manuscript. Most of these changes are editorial and minor in nature, with the intention to ensure the best clarity, accuracy, and flow as possible. All these changes are highlighted by using the Track Changes function. To make things easier, I have also provided a final clean copy of the final version of the manuscript accepting all changes I made using Track Change. This clean copy also contains the in-text citation using the format required by the journal.
Below I provide a detailed item-by-item response to each of the suggested changes you raised.
I would also like to share with you that GenBank accessions for complete or partial genome sequences for viruses identified through RNA-Seq are provided in the manuscript. We also reformatted both the in-text citation and the reference list according to the format required for viruses.
Once again, thank you so much for the time, effort and care you put in to conduct such a thorough review of our manuscript. Your input has helped us to produce a better paper when it is published.
Please let us know if you have any further questions or need assistance from us.
Sincerely yours,
Baozhong Meng
Professor of virology